# Menstrual hygiene management practices and associated health outcomes among school-going adolescents in rural Gambia

Helen M. Nabwera[1,2], Vishna Shah[1,3], Rowena Neville[3], Fatou Sosseh[3], Mariama Saidykhan[3], Fatou Faal[3], Bakary Sonko[3], Omar Keita[4], Wolf-Peter Schmidt[5], Belen Torondel[1]*

1 Environmental Health Group, Department of Infectious Diseases, London School of Hygiene and Tropical Medicine, London, United Kingdom, 2 Department of Clinical Sciences, Liverpool School of Tropical Medicine, Liverpool, United Kingdom, 3 Nutrition Theme, MRCG Keneba, Medical Research Council Unit, Banjul, The Gambia, 4 Regional Education Directorate Four, Ministry of Basic and Secondary Education, Mansakonko Lower River Region, The Gambia, 5 Department of Disease Control, London School of Hygiene and Tropical Medicine, London, United Kingdom

* Belen.Torondel@lshtm.ac.uk

**Data Availability Statement:** All relevant data are within the manuscript and its Supporting Information files.

## Abstract

Inadequate menstrual hygiene management (MHM) practices have been associated with adverse health outcomes. This study aimed to describe MHM practices among schoolgirls from rural Gambia and assess risk factors associated with urogenital infections and depressive symptoms. A cross-sectional study was conducted among adolescent schoolgirls in thirteen schools in rural Gambia. A questionnaire was used to collect information on socio-demographics, MHM practices and clinical symptoms of reproductive and urinary tract infections (UTIs). A modified Beck Depression Inventory-II was used to screen for depressive symptoms. Mid-stream urine samples were collected to assess for UTIs. Modified Poisson regression analysis was used to determine risk factors for symptoms of urogenital infections and depression among adolescent girls. Three hundred and fifty-eight girls were recruited. Although, 63% of the girls attended schools providing free disposable pads, reusable cloths/towels were the commonest absorbent materials used. Heavy menstrual bleeding was associated with depressive symptoms (adjusted prevalence ratio, aPR 1.4 [95% CI 1.0, 1.9]), while extreme menstrual pain (aPR 1.3 [95% CI 1.2, 1,4]), accessing sanitary pads in school (aPR 1.4 [95% CI 1.2, 1.5]) and less access to functional water source at school (aPR 1.4 [95% CI 1.3, 1.6]) were associated with UTI symptoms. Conversely, privacy in school toilets (aPR 0.6 [95% CI 0.5, 0.7]) was protective for UTI symptoms. Heavy menstrual bleeding (aPR 1.4 [95% CI 1.1, 2.0]) and taking <30 minutes to collect water at home were associated with RTI symptoms (aPR 1.2 [95% CI 1.0, 1.5]) while availability of soap in school toilets (aPR 0.6 [95% CI 0.5, 0.8] was protective for RTI symptoms. Interventions to ensure that schoolgirls have access to private sanitation facilities with water and soap both at school and at home could reduce UTI and RTI symptoms. More attention is also needed to support girls with heavy menstrual bleeding and pain symptoms.

**Funding:** BT awarded funding from the Medical Research Council (https://mrc.ukri.org/funding/), Grant number (MR/N027744/1). The funder and sponsor had no role in study design, data collection and analysis, decision to publish, or preparation of the manuscript.

**Competing interests:** The authors have declared that no competing interests exist.

## Introduction

Menstrual hygiene management (MHM) is now a key public health priority for women and adolescent girls globally [1, 2]. Poor menstrual hygiene practices among adolescent girls in impoverished communities in low- and middle-income countries (LMICs) is common [2]. The key barriers are inadequate knowledge of MHM and lack of access to appropriate menstrual absorbents due to poverty or cultural practices [3–6]. In addition, there are often challenges with accessing adequate and appropriate water, sanitation and hygiene (WASH) in schools due to the poor infrastructural planning of schools and limited resources [6, 7].

Inadequate MHM among adolescent girls in LMICs is associated with adverse health and social outcomes [5, 8–10]. This ultimately inhibits the empowerment of girls and women in these settings [11]. In Kenya, lack of access to menstrual absorbents among adolescent girls from deprived communities is associated with early pregnancies and sexually transmitted diseases due to the high-risk behaviours that they often adopt with older men [4, 8]. In India, the use of reusable absorbent material among non-pregnant women (>18y) attending outpatient clinics, was associated with a two-fold increase in urine and reproductive tract infections (including bacterial vaginosis and candida), while increased wealth and more space for personal hygiene were protective [12, 13]. Recent studies from Uganda and Indonesia have also shown that school absenteeism among adolescent girls is associated with inadequate MHM with an up to six-fold increased odds of school absenteeism during menstruation among Ugandan adolescents [10, 14]. MHM is also associated with feelings of shame, stress and anxiety among adolescents [3, 15–18], but there is limited data on the association with common mental health disorders, such as depression.

Knowledge of the association between different menstrual hygiene practices and health outcomes among adolescents in rural Gambia is not known. This limits the development of context relevant interventions. The objectives of this study were therefore 1) to describe menstrual hygiene practices and access to menstrual absorbents and WASH among school-going adolescent girl in a rural district in Gambia and 2) to assess associations of these practices with the risk of the three following outcomes: reproductive tract infections, urinary tract infections and depressive symptoms.

## Material and methods

### Study setting and population

This study was conducted in thirteen schools (seven English-based and six Arabic-based) within ten villages (i.e. Bajana, Janneh Kunda, Jiffarong, Kaiaf, Kantongkunda, Keneba, Kwinella, Nema Kuta, Nioro Jattaba and Sibito) in rural Kiang districts, in the lower river region of The Gambia [3]. The villages were within 10 miles of the Medical Research Council Unit The Gambia (MRCG) field station that provides free primary health care services and 24 h emergency care services [35]. There are also Ministry of Health primary health care clinics in the district.

The Gambia has two formal education systems: Arabic-based and English-based. The English-based schools are free public schools and the Arabic schools are private schools that focus on Quranic education [19]. The population is predominantly from the Mandinka ethnic group, the majority are Muslims and most families are polygamous [20]. Subsistence farming is the main income generating activity, but food insecurity is a recurring problem particularly in the wet season after a prolonged dry season (November- May) [21, 22]. The majority of the population live below the moderate poverty line of less than US$ 2/ day [21, 23].

## Study design and sampling

This was a school-based cross-sectional study that was preceded by a mixed methods study that explored the knowledge and perceptions of menstruation in this community [3]. Adolescent girls were recruited from thirteen schools (four English-based primary, three English-based secondary, six Arabic-based). Schools were selected based on the proximity to the MRCG field station in Keneba and the recommendation of the regional officials from the Ministry of Basic and Secondary Education.

Data presented in this paper are part of a cross-sectional study that aimed to explore the association between school absenteeism and MHM practices (described in other manuscript). In this paper we focus on the results of the association between MHM practices with health outcomes. The samples size calculation was based on estimates from our previous cross-sectional study [3] and estimates of school attendance and prevalence of urogenital infections described in other African Settings [24, 25], as we did not have estimates for these parameters in The Gambia. Using an estimated rate of prevalence of urogenital infection of 27%, use of reusable cloth of 40%, prevalence of drying inside the house of 21.6% and prevalence of school absenteeism of 23%. 40% prevalence was taken which gave us a larger sample size. Based on these estimates and using precision/absolute error of 5% and a type 1 error of 5%, a sample size of 368 was calculated.

All the schoolgirls aged 15–21 years were selected from the school registers provided by the teachers. Eligible participants were those girls had already had their menarche at the time of selection. Girls who were pregnant. This was ascertained by asking the girls if they were pregnant and those who said "yes" were excluded. Written informed consent was sought from students aged ≥18 years, and from the parents/caretakers of those aged <18 years, with student assent [26]. Participation was voluntary, and confidentiality was ensured by conducting interviews in a private area and keeping the data collected on secure servers without identifying information. All the field received training on how to take consent from the lead researcher using resources from the London School of Hygiene and Tropical Medicine (LSHTM) research governance.

## Data collection

Data were collected from the adolescent girls by two trained female field assistants in a private room on the school grounds. Each field assistant administered a standard pretested questionnaire to collect sociodemographic data and quantify knowledge and practices of menstruation (including age of menarche; knowledge and attitudes about menstruation; menstrual hygiene practices such as type of menstrual absorbent used, change frequency, washing and drying practices of reusable material; access to water, sanitation and hygiene at home; frequency of symptoms suggestive of urinary or reproductive tract infections). In addition, a modified Beck Depression Inventory II (BDI II) was used to screen for symptoms suggestive of depression among the adolescents.

Participants were requested to provide a mid-stream urine sample in a sterile specimen container. All specimen containers were labelled with participant's study ID. The urine samples were analysed by the lead researcher (who had been trained to conduct these tests by the clinic staff at MRC Keneba) in the school setting, using urine dipstick analysis with the Combur-test strips [27]. This is a quick screening test for a urinary tract infection (UTI) in children and adolescents, whose results are interpreted in the context of clinical feature suggestive of a UTI (and microscopy for evidence of bacteriuria where laboratory support is available) [28]. The guidance for urine collection that was given to the participants and the protocol for urine testing can be found in the S1 File. Urine samples were available for 352 (98%) of girls.

Participants whose urine dipstick tests were suggestive of a UTI (i.e. positive for both leucocyte esterase and nitrites or positive for only nitrites) or that had symptoms of UTI, were sent to the MRC Keneba clinic for further assessment and treatment.

Unannounced visits to conduct spot check of WASH facilities were conducted in each school by the team's supervisor. A pre-determined tool was used to assess the WASH hardware available; number and type of toilets, handwashing facilities and disposal facilities. As well as access the quality of these facilities; functionality, gender specificity, privacy, cleanliness, and availability of water and soap. Permissions to conduct unannounced visits were obtained from each school at the beginning of the study. Data completeness and consistency were reviewed at the end of each data collection day by the team's supervisor (VS).

**Pilot testing of tools.** In the development of the questionnaire, we used existing validated tools for assessing socio-economic status and WASH in this rural Gambian community [3]. These were complemented by additional questions from validated tools for assessing menstrual hygiene management among adolescents in other LMICs [12, 13]. BDI II has 21 items with a score range of 0–3 (0 = symptom absent) and (3 = severe symptom), with a total potential score of 63 [29]. It is a validated screening tool for symptoms suggestive of depression among adolescents in sub-Saharan Africa [29, 30]. A score of > 16 is suggestive of clinical depression [31] and among adolescents in Malawi had a specificity of >80% for depression [30]. An experienced team consisting of a senior research assistant (VS) and 2 field assistants translated and adapted the BDI-II to optimise its acceptability among adolescents in this rural Gambian community. They had extensive discussions about the meaning of each item and the corresponding phrases in Mandinka (the main local language in this community) to ensure that the equivalent cultural concepts were derived in a standardised manner. Forward and backward translation was done to ensure accuracy of the translations. We dropped one item that explore "interest in sex" from our study as it was not culturally appropriate for adolescent girls in this setting to discuss this. Our modified version therefore had 20 items (with a total potential score of 60) but we maintained the cut off score of >16. The questionnaire and BDI-II were first pilot tested among 4 female health care providers in rural Gambia, three of who were members of the local community. The purpose was to test translation and acceptability of questions and to support the training of field assistants. After incorporating the comments from this first phase, they were then pilot tested among three adolescent girls from the community to test feasibility, acceptability and quality of questions and responses. Feedback from this second phase was used to further amend the tools. All the tools were administered by the field assistants in the local language Mandinka.

## Data analysis

Data were collected using hard copy questionnaires that were double entered into SQL (SQL server 2017) and analysed using Stata (version 15.0, StataCorp, College Station, TX). Descriptive statistics using number and percentage for categorical data, and the mean and standard deviation (SD) or median and interquartile range (IQR) for continuous data, were used to summarise sociodemographic, menstrual hygiene practices and health outcome data. A principal component analysis (PCA) was used to determine household socioeconomic status using an asset based index in which we included the following fourteen socioeconomic indicators: ownership of television, radio, cooker/stove, refrigerator, sofa/couch, boat, car/truck, electricity, motorcycle, bicycle, animal cart, mobile phone (all binary) and type of walls and flooring in the family house [32]. The households that the adolescent girls belonged to were classified into three quantiles (i.e. Poorest/2nd quantile/least poor) based on Filmer and Pritchett's method [33].

Guidelines on latrine to student ratio vary widely. WHO recommendations for latrine to female student ratio is 1:25 [34]. The WASH in Schools (WinS) standard for Eastern and Southern Africa Recommendations (ESAR) suggests 20–50 students per latrine [35]. Since no national guidelines for Gambia were found, the wider range of WinS standard for ESAR was used as a comparison for the analysis.

A participant was classified as having at least one symptom suggestive of a UTI if they had any of the following symptoms in the preceding 24 hours: feeling of burning or discomfort when urinating, passing urine more than once at night, having cloudy urine or blood in your urine, foul-smelling urine or having lower abdominal or vaginal pain. A urine dipstick test was classified as positive i.e. suggestive of a UTI if it was positive for both leucocyte esterase and nitrites or positive for nitrites alone [36, 37]. A participant was classified as having at least one symptom suggestive of a reproductive tract infection (RTI) if they had any of the following symptoms in the preceding 2 months: abnormal vaginal discharge, foul-smelling/fishy smell from genital area, or feeling of burning or itching in the genitalia. Girls who reported any of the above symptoms on the day the study team visited the schools were referred to either the MRCG field station clinic or a government primary health care clinic (depending on the proximity from the school).

Univariable and multivariable Poisson regression analyses of binary outcome variables were applied to provide both unadjusted and adjusted prevalence ratios in exploring factors associated with poor MHM and adverse health outcomes. Confidence intervals were adjusted for the binary nature of the data using robust standard errors [38]. In the multivariable Poisson regression analysis, we used three models, one for each of the health outcomes of interest (i.e. depression, UTI, RTI). For each model we included explanatory variables that were statistically significant in the univariable analysis (Table 5) as well as those that had biologically plausible associations with each of the three health outcomes [39]. The details of the selected variables can be found in Table 6. The final models for each health outcome had between 12–14 explanatory variables. Models that included school level variables were adjusted for clustering at school level using robust standard errors.

The sociodemographic variables included in the models had up to 19% of the data missing and the menstrual hygiene management modules had up to 14% of data missing. Missing data were not imputed. Adjusted prevalence ratios with 95% confidence intervals (CI) using a P-value< 0.05 were considered to be statistically significant associations with the outcome variable.

## Ethical approval

Ethics approval for the study was granted by the London School of Hygiene and Tropical Medicine, UK [Ref: 10225] and by the Gambia Government/Medical Research Council Joint ethics committee [SCC1426 and SCC1509]. Permission to conduct the study in the schools was obtained from the Ministry of Basic and Secondary Education, The Gambia.

## Results

### Characteristics of study participants

From November 2016 and July 2017, 358 (94% of eligible) adolescent girls attending the thirteen schools were recruited into the study (**Fig 1**). Their mean age was 16.9 years (Standard deviation, SD: 1.6, age range 15–21 years), all of them were Muslims. Six percent (17) were married, their mean age was 18.3 year (SD 1.8, range 15–21 years), 6 (35%) of them were < 18 years and <1% of the girls reported that they had at least one child (2/290). The majority walked to school (299, 83%) and for over a third the journey to school was < 15 minutes (139,

39%). Nearly half of them reported that their mothers had not received any formal education (143, 45%). Arabic school education was the predominant form of education reported for both fathers (184, 60%) and mothers (121, 38%) of the students. Most of adolescent girls came from families whose main source of income was farming (216, 60%) (**Table 1**).

**Table 1. Socio-demographic characteristics of schoolgirls participating in the study.**

| Characteristics | N = 358 |
|---|---|
| Age in years, mean (SD)[+] | 16.9 (1.6) |
| Muslim, n (%) | 358 (100) |
| Married, n (%)[*] | 17 (6) |
| Do not have children, n (%)[**] | 288 (99) |
| Means of getting to school, n (%) | |
| Walking | 299 (83) |
| Cycling | 56 (16) |
| Other | 2 (1) |
| Time it takes from home to school, n (%) | |
| <15minutes | 139 (39) |
| 15-30minutes | 107 (30) |
| 30-60minutes | 75 (21) |
| >60 minutes | 37 (10) |
| Education level of mother, n (%)[***] | |
| No formal education | 143 (45) |
| Some primary school | 19 (6) |
| Completed 6 years primary school | 20 (6) |
| Some secondary | 11 (3) |
| Completed secondary school or higher-level education | 4 (2) |
| Arabic school only | 121 (38) |
| Education level of father, n (%)[#] | |
| No formal education | 78 (25) |
| Some primary school | 3 (1) |
| Completed 6 years primary school | 8 (3) |
| Some secondary | 4 (1) |
| Completed secondary school or higher-level education | 32 (10) |
| Arabic school only | 184 (60) |
| Material of wall of house, n (%) | |
| Mud | 255 (71) |
| Cement | 103 (29) |
| Material of floor of house, n (%) | |
| Mud | 91 (25) |
| Cement | 263 (73) |
| Tiles | 4 (1) |
| Assets in household, n (%) | |
| Radio | 266 (74) |
| Television | 46 (13) |
| Car or truck | 40 (11) |
| Motorbike | 52 (15) |
| Bicycle | 298 (83) |
| Mobile phone | 357 (99.7) |

(*Continued*)

**Table 1.** (Continued)

| Characteristics | N = 358 |
|---|---|
| Electricity (generator/solar) | 171 (48) |
| Animal cart | 227 (63) |
| Gas cooker/stove | 74 (21) |
| Household head's main source of income, n (%) | |
| Farming | 216 (60) |
| Business | 30 (8) |
| Salary | 74 (21) |
| Trading | 27 (8) |
| Other | 11 (3) |
| Socioeconomic quantiles | |
| Poorest | 133 (37) |
| 2nd quantile | 106 (30) |
| Least poor | 119 (33) |

SD- Standard Deviation

* 67 unknown

** 67 unknown

*** 40 unknown

# 49 unknown

+ Age range: 15–21 years

## Access to water, hygiene and sanitation

Regarding access to water, hygiene and sanitation, the main source of household water for the girls was a community standpipe (235, 66%) and nearly half of the girls reported that it took them less than 15 minutes to and from their household water source (185, 52%). Pit latrines with a slab were the commonest form of household sanitation (301, 84%) (**Table 2**).

**Table 2. Access to water and sanitation at home and school.**

| Access to water and sanitation | N = 358 |
|---|---|
| *Access at home (self-reported data)* | |
| Main source of water for household over preceding 6 months, n (%) | |
| Household standpipe or handpump | 3 (<1) |
| Community standpipe | 235 (66) |
| Community handpump | 85 (24) |
| Protected well | 6 (2) |
| Unprotected well | 29 (8) |
| Time taken to collect water (to and from source), n (%) | |
| <15minutes | 185 (52) |
| 15-29minutes | 99 (28) |
| 30-59minutes | 65 (18) |
| 1-3hrs | 9 (2) |
| Main toilet facility in household, n (%) | |
| Flush toilet | 3 (<1) |
| Pour flush toilet | 12 (3) |
| Pit latrine with a slab | 301 (84) |
| Pit latrine without a slab | 39 (11) |

(*Continued*)

**Table 2.** (Continued)

| Access to water and sanitation | N = 358 |
|---|---|
| Use the bush | 1 (<1) |
| Another household's facility | 2 (<1) |
| ***Access at school (spot check data)*** | |
| Main source of water in schools, n (%) | |
| School standpipe or handpump | 338 (94) |
| Community standpipe | 20 (6) |
| How often is the water source functional? n (%) | |
| 5–7 days per week | 325 (91) |
| 2–4 days per week | 33 (9) |
| Fewer than 2 days per week | 0 |
| Water source functional on the visit day to the school, n (%) | 253 (71) |
| Duration of time when water sources were not functional, n (%) | |
| Less than one day | 34 (32) |
| More than one day and less than one week | 54 (52) |
| More than one week and less than one month | 17 (16) |
| More than one month | 0 |
| School had handwashing facilities, n (%) | 358 (100) |
| Type of handwashing facilities in schools, n (%) | |
| Running water from a piped system or tank | 158 (44) |
| Hand-poured water system (bucket or ladle) | 314 (88) |
| Basin/bucket (not poured) | 0 |
| Other | 45 (13) |
| Location of handwashing facilities, n (%)* | |
| Inside toilet block | 49 (14) |
| Outside/Away from toilet block | 309 (86) |
| Availability of soap for handwashing, n (%) | 45 (13) |
| School toilet facilities, n (%) | |
| Meet the WASH in Schools in the Eastern and Southern Africa Region latrine: student ratio** | 147 (41) |
| Clean*** | 317 (89) |
| Privacy | |
| Door | 315 (88) |
| No door | 43 (12) |
| Water inside the toilet cubicle for cleaning, n (%) | |
| No water in any toilet cubicle | 272 (76) |
| Water in at least one cubicle | 86 (24) |
| Bins available for disposing used menstrual blood absorbents | 0 |
| Pit/Incinerator for burning absorbents available | 0 |

*More than one answer could be selected therefore n>358

**WASH in Schools standard for Eastern and Southern Africa Recommendations– 20–50 students per latrine

***Clean toilet was defined as one with no visible faecal matter or urine in or around the cubicle, no litter or dirt in the cubicle.

In school, the main source of water for most of the girls was a school standpipe or hand pump (338, 94%), most of which were reported to be functional 5–7 days of the week (325, 91%). At the time of the school spot check visit, the study team found that 253 girls (71%) had access to water. All the girls attended to schools that had some type of handwashing facilities,

the majority had a hand-poured water system (bucket or ladle) (314, 88%) with no soap (308, 87%) and were away from the toilets (309, 86%).

All schools had improved toilet facilities; either pit latrine with a slab or pour flush toilets. Most of the schools fit the basic sanitation criteria, having improved facilities, single sex and usable at the time of visit. Only one school (an Arabic school) did not have gender specific toilets. The latrine to student ratio for girls ranged from 1:26 to 1:125 and that from boys ranged from 1:29 to 1:171. None of the schools met the WHO latrine to female student ratio recommendations. Forty-one percent (147) of girls attended a school that met the WinS standard for ESAR. Most of the girls attended schools that had clean toilets (317, 89%) and privacy; a door (315, 88%) or a door that is lockable (282, 79%). Whereas less than a quarter (86, 24%) of girls went to schools that had water inside at least one of the cubicles (**Table 2**).

None of the schools had bins for disposing used menstrual blood absorbents or a pit/incinerator for burning absorbents (**Table 2**).

## Menstrual hygiene practices

The mean age at their first menstrual period (menarche) was 14.4 years (SD 1.2, age range 10–18). Over half of the girls reported having "heavy menstrual bleeding" (191, 53%) in the preceding two months (191, 53%), while over a third suffered extreme pain during their menstrual periods (137, 38%). Irregular bleeding was infrequent. Most of the girls reported that they had learnt about menstruation before menarche (228, 64%). Almost all them reported that disposable sanitary pads (311, 87%) and reusable cloths/towels (355, 99%) could be used to manage menstrual blood. However, they had no knowledge of menstrual hygiene cups or tampons. Reusable cloths/towels were the commonest absorbent materials used by the girls (192, 53%) and the majority reported that they used soap and water to wash their absorbent material (302, 98%) and dried them in the "bathroom" at home (264, 85%). Less than half of girls used disposable pads (165, 46%), although nearly two thirds attended schools that had a supply of disposable pads for the girls (208, 63%). The mean number of packets of disposable pads provided to each girl last time they were to collect them was 1.3, SD 0.6 (each packet had 10 pads). Nearly a third of the girls (101, 28%) reported that they would like to purchase disposable sanitary pads themselves from the local village shops but were not able to, the commonest barriers were insufficient funds (48, 48%) and embarrassment (45, 45%). The majority reported that during their menstrual period they changed their absorbent material ≥3 time a day (200, 56%) and had a bath including vaginal wash at least twice a day (339, 95%) (**Table 3**).

**Table 3. Menstrual hygiene management practices of schoolgirls participating in the study.**

| Menstrual hygiene practices | N = 358 |
|---|---|
| Age at menarche in years (first menstrual period), mean (SD) | 14.4 (1.2) |
| Learnt about menstruation before menarche, n (%) | 228 (64) |
| **Difficulties with menstruation** | |
| Bleeding between menses in the preceding 2 months, n (%) | 7 (2) |
| Heavy menstrual bleeding in the preceding 2 months, n (%) | 191 (53) |
| Extreme level of pain experienced during menstrual period, n (%) | 137 (38) |
| Mild level of pain experienced during menstrual period, n (%) | 162 (45%) |
| Knowledge of absorbent material for menstrual blood, n (%) | |
| Disposable sanitary pads | 311 (87) |

(*Continued*)

**Table 3.** (Continued)

| Menstrual hygiene practices | N = 358 |
|---|---|
| Reusable cloths/towels | 355 (99) |
| Tampon/Menstrual cup/Other | 0 |
| Absorbent materials used during menstrual period, n (%) | |
| Disposable sanitary pads | 165 (46) |
| Reusable cloth/towel | 192 (53) |
| Knickers | 1 (<1) |
| Would have liked to purchase disposable pads from the local shops but were unable to, n (%)* | 101 (28) |
| Reasons for not being able to purchase disposable pads from local shop, n (%)¥ | |
| *"I didn't have enough money"* | 48 (48) |
| *"Not available in shops"* | 18 (18) |
| *"I felt embarrassed"* | 45 (45) |
| Attend school that has a supply of pads for girls, n (%)** | 208 (63) |
| Number of packets of disposable sanitary pads provided by the school last time they collected, mean (SD) ¥¥ | 1.3 (0.6) |
| Attend school that has emergency supply of pads, n (%)*** | 190 (61) |
| Frequency of changing absorbent material on one of the heavy bleeding days, n (%) | |
| Once a day | 10 (3) |
| Twice a day | 148 (41) |
| Three or more times a day | 200 (56) |
| Location for changing absorbent material at home for the girls, n (%)* | |
| Outside bathroom | 346 (97) |
| Private room | 11 (3) |
| How the absorbent material is washed at home, n (%)# | |
| With water | 7 (2) |
| With water and soap/detergent | 302 (98) |
| Other | 1 (<1) |
| Mode of drying washed absorbent material at home, n (%) ## | |
| In the bathroom | 264 (85) |
| In the sun/open space | 1 (<1) |
| Inside the house/room | 43 (14) |
| Under the mattress | 1 (<1) |
| Frequency of bathing including vaginal wash during menstruation, n (%) | |
| Once a day | 19 (5) |
| Twice a day or more | 339 (95) |

SD- standard deviation

*1 missing data

¥Each girl could have more than one response

**27 not known

¥¥ One packet contained 10 disposable sanitary pads

***47 do not know

#48 not known

##49 not known

## Health status of the adolescent girls

The median modified BDI II score was 11 (IQR 7, 15) and 74 (21%) of the adolescent girls had symptoms suggestive of depression (i.e. score>16). Although the majority of girls (269, 75%) had

at least one symptom suggestive of a UTI in the preceding 24 hours, only 3 (<1%) girls had a urine dipstick test suggestive of a UTI. Two of the girls with at least one symptom suggestive of UTI had a urine dipstick positive for nitrites. Only one girl with at least one symptom suggestive of UTI had a urine dipstick positive for both leucocyte esterase and nitrites. Forty-two (12%) girls had urine dipstick tests that were leucocyte esterase positive suggesting an infection or inflammation in the lower urogenital tract (not a UTI). Twenty-three (7%) girls with symptoms suggestive of RTI had a leucocyte esterase positive urine dipstick. Ten (3%) girls had urine dipstick tests that were positive for blood. Half of them also had symptoms suggestive of RTI.

Less than half (168, 47%) of the girls had at least one symptom suggestive of an RTI in the preceding 2 months. Lower abdominal or vaginal pain was the commonest symptom reported (221, 62%). In addition, symptoms suggestive of genital irritation including dysuria (82, 23%) and burning or itching of the genitalia (161, 45%) were reported by nearly a quarter of the girls. Abnormal vaginal discharge was also reported by about a quarter of the girls (79,22%) (**Table 4**).

## Association between menstrual hygiene practices and health status of adolescent girls

**Depression.** In the univariable regression analysis age ≥18 years (unadjusted PR 2.1 [95% CI 1.4, 3.2]) and extreme pain during menstruation (unadjusted PR 1.9 [95%CI 1.3, 2.8]) were

**Table 4. Prevalence of symptoms and health outcomes among schoolgirls participants.**

| Health status | N = 358 |
| --- | --- |
| **Mental health** | |
| **Depression**[*] | |
| Modified Beck Depression Inventory II[1,2], median (IQR) | 11 (7, 15) |
| Depression, n (%) | **74 (21)** |
| **Physical health** | |
| **Urinary Tract Infection Symptoms** | |
| Feeling of burning or discomfort when urinating (dysuria) in the preceding 24 hours, n (%) | 82 (23) |
| Passing urine more than once at night (urinary frequency) in the preceding 24 hours, n (%) | 56 (16) |
| Cloudy urine or blood in urine in the preceding 24 hours, n (%) | 86 (24) |
| Foul-smelling urine in the preceding 24 hours, n (%) | 64 (18) |
| Lower abdominal or vaginal pain | 221 (62) |
| **At least one symptom suggestive of urinary tract infection in the preceding 24hr, n(%)** | **269 (75)** |
| **Urine dipstick results**[**] | |
| Leucocyte esterase positive, n (%) | 42 (12) |
| Nitrite positive, n (%) | 3 (<1) |
| Blood positive, n(%) | 10 (3) |
| **Urine dipstick results suggestive of UTI, n (%)** | 3 (<1) |
| **Reproductive Tract Infection Symptoms** | |
| Abnormal vaginal discharge | 79 (22) |
| Foul-smelling/fishy smell from genital area | 61 (17) |
| Burning or itching of genitalia | 161 (45) |
| **At least one symptom suggestive of reproductive tract infection in the preceding 2 months, n (%)** | **168 (47)** |

[1]Beck AT, Steer RA, Garbin MG. Psychometric properties of the Beck Depression Inventory: Twenty-Five years of evaluation. Clin Psychol Rev 1988; 8:77–100.

[2]Beck AT, Steer RA, Brown GK. BDI-II: Beck Depression Inventory Manual. 2nd Ed. San Antonio, TX: Psychological Corporation; 1996.

[*]Depression screening using modified Depression Inventory II, Depression if score >16

[**] **Collected in 352 participants (98%)**

associated with an increased prevalence ratio of symptoms suggestive of depression. The time taken to collect water <30 minutes at home was associated with reduced prevalence ratio of symptoms suggestive of depression (unadjusted PR 0.4 [95%CI 0.2, 0.9]) (**Table 5**).

In the multivariable regression analysis only heavy menstrual bleeding (aPR 1.4 [95% CI 1.0, 1.9]) was associated with increased prevalence ratio of symptoms suggestive of depression (**Table 6**).

**Urinary tract infections.** In the univariable regression analysis, heavy menstrual bleeding (unadjusted PR 1.3 [95% CI 1.2, 1.5]), extreme pain during menstruation (unadjusted PR 1.4 [95%CI 1.3, 1.6]), accessing sanitary pads in school (unadjusted PR 1.1 [95% CI 1.0, 1.3]) and water source in school being functional for only 2–4 days per week (unadjusted PR 1.1 [95% CI 1.0, 1.3]) were associated with increased prevalence ratio of UTI symptoms in the previous 24 hours. Conversely, access to clean toilets (unadjusted PR 0.9 [95%CI 0.8, 1.0]) and privacy in the school toilets (unadjusted PR 0.9 [95%CI 0.8, 1.0]) were associated with reduced prevalence ratio of UTI symptoms in the previous 24 hours (**Table 5**).

In the multivariable analysis, extreme pain during menstruation (aPR 1.3 [95% CI 1.2, 1.4]), accessing sanitary pads in school (aPR 1.4 [95% CI 1.2, 1.5]) and attending schools with water source that was functional only 2–4 days per week (aPR 1.4 [95% CI 1.3, 1.6]) were associated with increased prevalence ratio of at least one UTI symptom in the previous 24 hours. Privacy in the school toilets (aPR 0.6 [95% CI 0.5, 0.7]) was associated with decreased prevalence ratio of UTI symptoms (**Table 6**).

**Reproductive tract infections.** In the univariable analysis, heavy menstrual bleeding (unadjusted PR 1.5 [95% CI 1.2, 1.8]) and extreme pain during menstruation (unadjusted PR 1.4 [95%CI 1.1, 1.6]) were associated with increased prevalence ratio of at least one RTI in the last 2 months; whilst accessing clean school toilets (unadjusted PR 0.7 [95% CI 0.5, 0.8]) and privacy in the school toilets (unadjusted PR 0.8 [95% CI 0.7, 1.0]) were associated with decreased prevalence ratio of RTI symptoms (**Table 5**).

In the multivariable analysis, heavy menstrual bleeding (aPR 1.4 [95% CI 1.1, 1.8]) and duration<30 minutes to collect water at home (aPR 1.2 [95% CI 1.0, 1.5]) were associated with increased prevalence ratio of at least one RTI symptom in the previous 2 months. Availability of soap to wash hands in the school toilets (aPR 0.6 [95% CI:0.5, 0.8]) were associated with decreased prevalence ratio of RTI symptoms (**Table 6**).

## Discussion

Our study found that reusable cloths/towels were the commonest absorbent materials used by the girls. Less than half of girls used disposable pads, although nearly two thirds attended schools that distributed free disposable pads. The type of absorbent material used was surprisingly not associated with any of the health outcomes described in this study (i.e. depression, UTI or RTI). UTI and RTI symptoms were common and associated with sub-optimal access to water, sanitation and hygiene facilities at school. UTI symptoms were also associated with extreme menstrual pain and accessing disposable sanitary pads in school while RTI symptoms were also associated with heavy menstrual bleeding and duration<30 minutes to collect water at home. Symptoms suggestive of depression were also common and associated with heavy menstrual bleeding.

We found that majority of the school-going adolescent girls reported having access to a reliable water source and basic sanitation at home. However, there were key deficiencies in the provision of WASH in some of the schools—a pre-requisite for appropriate MHM [40], including lack of privacy where toilets had no doors (12%), dirty toilets (11%), water sources that were only function for only 2–4 days per week (9%), water far from the cubicles (76%),

**Table 5. Factors associated with adverse health outcomes (Univariable analysis).**

| Exposure variables | (Unadjusted) Incidence Rate Ratio (95% CI) | P value |
|---|---|---|
| *Depression* | | |
| Age | | |
| 17 years or younger | 1.0 | |
| **18 years or older** | **2.1 (1.4, 3.2)** | **<0.001** |
| Marital status | | |
| Not married | 1.0 | 0.06 |
| Married | 2.0 (1.0, 3.9) | |
| Paternal education | | |
| No formal education | 1.0 | |
| Arabic education only | 1.0 (0.6, 1.6) | 0.91 |
| Primary level or more | 0.7 (0.3, 1.5) | 0.35 |
| Maternal education | | |
| No formal education | 1.0 | |
| Arabic education only | 0.8 (0.5, 1.3) | 0.38 |
| Primary level or more | 1.1 (0.6, 2.0) | 0.72 |
| Water collection time | | |
| 30 minutes or more | 1.0 | |
| **Less than 30 minutes** | **0.5 (0.2, 0.9)** | **0.03** |
| Time when they learn about menstruation | | |
| Learn about menstruation pre menarche | 1.0 | |
| Learn about menstruation post menarche | 1.1 (0.7, 1.7) | 0.56 |
| Provision of pads at school | | |
| No supply of school pads | 1.0 | |
| School providing girls with a supply of pads | 0.7 (0.5, 1.1) | 0.11 |
| Frequency of changing pads | | |
| Changing pads once a day | 1.0 | |
| Change twice a day | 1.8 (0.3, 11.7) | 0.56 |
| Change three or more times a day | 2.4 (0.4, 15.4) | 0.37 |
| Material normally used during the girls' monthly period | | |
| Reusable cloth | 1.0 | |
| Disposable pads | 0.8 (0.5, 1.2) | 0.32 |
| After washing it, how is it dried? | | |
| Inside house or under mattress | 1.0 | |
| Open space or bathroom | 0.7 (0.4, 1.2) | 0.23 |
| How often water source in school is functional | | |
| 5–7 days per week | 1.0 | |
| 2–4 days per week | 1.4 (0.5, 3.4) | 0.5 |
| Availability of soap in school for handwashing | | |
| No | 1.0 | |
| Yes | 0.4 (0.04, 3.7) | 0.4 |
| Location of handwashing facility in school | | |
| Away from toilet block | 1.0 | |
| Inside toilet block | 0.8 (4.2, 1.38) | 0.37 |
| Water in at least one toilet cubicle at school | | |
| No | 1.0 | |
| Yes | 1.2 (0.5, 2.7) | 0.7 |

*(Continued)*

**Table 5.** (Continued)

| Exposure variables | (Unadjusted) Incidence Rate Ratio (95% CI) | P value |
|---|:---:|:---:|
| Clean Toilets at school | | |
| No | 1.0 | |
| Yes | 1.5 (0.9, 2.4) | 0.13 |
| Privacy in the toilets at school | | |
| No door | 1.0 | |
| Door in place | 10.0 (0.8, 121.51) | 0.07 |
| School toilets meet WASH in Schools in the Eastern and Southern Africa Region standards | | |
| No | 1.0 | |
| Yes | 0.9 (0.4,2.3) | 0.87 |
| Menstrual bleeding flow | | |
| No heavy menstrual bleeding | 1.0 | |
| Heavy menstrual bleeding | 1.5 (1.0, 2.3) | 0.05 |
| Menstrual pain | | |
| No pain or mild pain | 1.0 | |
| **Extreme pain** | **1.9 (1.3, 2.8)** | **0.002** |
| Socioeconomic quantiles | | |
| Poorest | 1.0 | |
| 2nd quantile | 0.8 (0.5, 1.4) | 0.46 |
| Least poor | 1.0 (0.6, 1.6) | 0.99 |
| *Urinary Tract Infections* | | |
| Age | | |
| 17 years or younger | 1.0 | |
| 18 years or older | 1.08 (0.96, 1.22) | 0.22 |
| Marital status | | |
| Not married | 1.0 | 0.13 |
| Married | 1.2 (1.0, 1.4) | |
| Paternal education | | |
| No formal education | 1.0 | |
| Arabic education only | 1.0 (0.9, 1.2) | 0.87 |
| Primary level or more | 1.0 (0.8, 1.2) | 0.90 |
| Maternal education | | |
| No formal education | 1.0 | |
| Arabic education only | 1.0 (0.8, 1.1) | 0.48 |
| Primary level or more | 0.9 (0.8, 1.1) | 0.42 |
| Water collection time at home | | |
| 30 minutes or more | 1.0 | |
| Less than 30 minutes | 1.0 (0.9, 1.2) | 0.67 |
| Time when they learn about menstruation | | |
| Learn about menstruation premenarche | 1.0 | |
| Learn about menstruation post menarche | 1.0 (0.9, 1.2) | 0.39 |
| Provision of sanitary pads in school | | |
| No supply of school pads | 1.0 | |
| **School providing girls with a supply of pads** | **1.1 (1.0, 1.3)** | **0.04** |
| Frequency of changing pads | | |
| Changing pads once a day | 1.0 | |

(*Continued*)

**Table 5.** (Continued)

| Exposure variables | (Unadjusted) Incidence Rate Ratio (95% CI) | P value |
|---|---|---|
| Change twice a day | 1.8 (0.3, 11.7) | 0.56 |
| Change three or more times a day | 2.4 (0.4, 15.4) | 0.37 |
| Material normally used during the girls' monthly period | | |
| Reusable cloth | 1.0 | |
| Disposable pads | 1.1 (1.0, 1.2) | 0.21 |
| After washing it, how is it dried? | | |
| Inside house or under mattress | 1.0 | |
| Open space or bathroom | 1.1 (0.9, 1.3) | 0.54 |
| How often water source in school is functional | | |
| 5–7 days per week | 1.0 | |
| **2–4 days per week** | **1.1 (1.0, 1.3)** | **0.03** |
| Availability of soap in school for handwashing | | |
| No | 1.0 | |
| Yes | 0.9 (0.7, 1.2) | 0.47 |
| Location of handwashing facility in school | | |
| Away from toilet block | 1.0 | |
| Inside toilet block | 1.1 (0.9, 1.2) | 0.27 |
| Water in at least one cubicle at school | | |
| No | 1.0 | |
| Yes | 1.0 (0.9, 1.2) | 0.76 |
| Clean toilets at school | | |
| No | 1.0 | |
| **Yes** | **0.9 (0.8, 1.0)** | **0.01** |
| Privacy in the toilets at school | | |
| No door | 1.0 | |
| **Door in place** | **0.9 (0.8, 1.0)** | **0.01** |
| School toilets meet school toilets in schools in the Eastern and Southern Africa Region standards | | |
| No | 1.0 | |
| Yes | 0.9 (0.8, 1.1) | 0.48 |
| Menstrual bleeding flow | | |
| No heavy menstrual bleeding | 1.0 | |
| **Heavy menstrual bleeding** | **1.3 (1.2, 1.5)** | **<0.005** |
| Menstrual pain | | |
| No pain or mild pain | 1.0 | |
| **Extreme pain** | **1.4 (1.3, 1.6)** | **<0.005** |
| Socioeconomic quantiles | | |
| Poorest | 1.0 | |
| 2nd quantile | 1.1 (1.0, 1.3) | 0.17 |
| Least poor | 1.1 (0.9, 1.3) | 0.23 |
| *Reproductive Tract Infections* | | |
| Age | | |
| 17 years or younger | 1.0 | |
| 18 years or older | 1.1 (0.9, 1.4) | 0.17 |
| Marital status | | |
| Not married | 1.0 | 0.58 |

(*Continued*)

**Table 5.** (Continued)

| Exposure variables | (Unadjusted) Incidence Rate Ratio (95% CI) | P value |
|---|---|---|
| Married | 1.1 (0.8, 1.6) | |
| Paternal education | | |
| No formal education | 1.0 | |
| Arabic education only | 0.9 (0.7, 1.2) | 0.54 |
| Primary level or more | 1.0 (0.7, 1.3) | 0.87 |
| Maternal education | | |
| No formal education | 1.0 | |
| Arabic education only | 1.0 (0.8, 1.3) | 0.81 |
| Primary level or more | 1.0 (0.7, 1.3) | 0.76 |
| Water collection time at home | | |
| 30 minutes or more | 1.0 | |
| Less than 30 minutes | 1.2 (1.0, 1.4) | 0.08 |
| Time when they learn about menstruation | | |
| Learn about menstruation pre menarche | 1.0 | |
| Learn about menstruation post menarche | 1.1 (0.9, 1.4) | 0.26 |
| Provision of sanitary pads in school | | |
| No supply of school pads | 1.0 | |
| School providing girls with a supply of pads | 1.0 (0.9, 1.2) | 0.80 |
| Frequency of changing pads | | |
| Changing pads once a day | 1.0 | |
| Change twice a day | 1.0 (0.6, 1.7) | 0.99 |
| Change three or more times a day | 0.9 (0.6, 1.6) | 0.82 |
| Material normally used during the girls' monthly period | | |
| Reusable cloth | 1.0 | |
| Disposable pads | 1.0 (0.8, 1.1) | 0.65 |
| After washing it, how is it dried? | | |
| Inside house or under mattress | 1.0 | |
| Open space or bathroom | 1.0 (0.9, 1.2) | 0.90 |
| How often water source in school is functional | | |
| 5–7 days per week | 1.0 | |
| 2–4 days per week | 1.0 (0.9, 1.1) | 0.87 |
| Availability of soap in school for handwashing | | |
| No | 1.0 | |
| Yes | 0.9 (0.7, 1.2) | 0.47 |
| Location of handwashing facility in school | | |
| Away from toilet block | 1.0 | |
| Inside toilet block | 1.1 (0.9, 1.2) | 0.45 |
| Clean toilets at school | | |
| No | 1.0 | |
| **Yes** | **0.8 (0.7, 0.9)** | **0.001** |
| Privacy in the toilets at school | | |
| No Door | 1.0 | |
| **Door in place** | **0.8 (0.7, 1.0)** | **0.016** |
| Water inside at least one cubicle at school | | |
| No | 1.0 | |
| Yes | 1.1 (0.9, 1.2) | 0.5 |

(*Continued*)

**Table 5.** (Continued)

| Exposure variables | (Unadjusted) Incidence Rate Ratio (95% CI) | P value |
|---|---|---|
| School toilets meet WASH in schools in the Eastern and Southern Africa Region standards | | |
| No | 1.0 | |
| Yes | 0.8 (0.7, 1.0) | 0.11 |
| Menstrual bleeding flow | | |
| No heavy menstrual bleeding | 1.0 | |
| **Heavy menstrual bleeding** | **1.5 (1.2, 1.8)** | **<0.005** |
| Menstrual pain | | |
| No pain or mild pain | 1.0 | |
| **Extreme pain** | **1.4 (1.1, 1.6)** | **<0.005** |
| Socioeconomic quantiles | | |
| Poorest | 1.0 | |
| 2[nd] quantile | 1.0 (0.7, 1.3) | 0.81 |
| Least poor | 1.1 (0.9, 1.4) | 0.47 |

*Only 4 (tertiary level maternal education)

absence of hand-washing facilities and soap for in the toilets (87%) and complete lack of absorbent disposal facilities. Indeed, girls who attended schools where the access to water and privacy of toilets were optimal had reduced prevalence ratios for UTI symptoms, while availability of soap to wash hands in school toilets was associated with reduced prevalence ratios for RTI symptoms, suggesting a protective effect. Also, less than half of the schools met the recommended latrine: student ratio of 20–50 students per latrine that meant that would have had implications for maintaining the cleanliness whilst girls would have been hindered from accessing the toilets during their menstrual period due to overcrowding that would have had a negative impact on their MHM practices during school time. These inadequacies are common particularly in rural schools in LMICs and lead to poor MHM practices [2, 41–43]. In rural Zambia, access to WASH in schools was a key challenge for school going adolescent girls, in particular toilets not having soap and water, poorly maintained toilets with no locks and having a bad odour [9]. In rural Uganda, girls reported having to go home from school to change their menstrual absorbents as well as wash and dry their reusable absorbents due to the inadequate WASH facilities in school, that has implications on their MHM but also educational attainment [15]. Our study showed that inadequate WASH facilities in schools hindered them from meeting the thresholds of appropriate MHM to avert adverse health outcomes, particularly for girls with heavy menstrual bleeding. Although this study cannot infer causation, but it opens new questions about the relevance of girls having a secure and comfortable place where they can change without stress and have better proximity to water, hand-washing facilities and soap. These findings have implications for future adolescent health strategies in schools in the Gambia and other similar LMICs context. Improving WASH in communities and at schools to prevent water and sanitation-related diseases among staff and children in schools was a key goal of The Gambia National Strategy for Sanitation and Hygiene 2011–2016 [44].

Duration of less than30 minutes to collect water at home was weakly associated with RTIs. Previous studies have shown that poor MHM practices are associated with RTIs and that access to adequate WASH would mitigate this [12]. Walking shorter distances (less than 7 min) to access water for bathing has been associated with fewer RTIs among women in rural

**Table 6. Factors associated with adverse health outcomes (Multivariable analysis).**

| Exposure variables | Adjusted Incidence Rate Ratio (95% CI) | P value |
|---|---|---|
| *Depression* | | |
| Age≥18y | 2.0 (1.0, 4.0) | 0.06 |
| Married | 1.3 (0.7, 2.2) | 0.39 |
| Maternal education | 0.9 (0.9, 1.0) | 0.11 |
| Less than 30 minutes to collect water at home | 0.5 (0.2, 1.3) | 0.16 |
| Learn about menstruation post menarche | 0.9 (0.6, 1.4) | 0.77 |
| School providing girls with a supply of pads | 0.6 (0.3, 1.3) | 0.21 |
| Frequency of changing pads (reference once a day) | 1.4 (1.0, 2.0) | 0.07 |
| Using other sanitary material other than disposable pads | 0.8 (0.5, 1.4) | 0.51 |
| Water source in school functional 2–4 days per week | 1.3 (0.5, 3.4) | 0.59 |
| **Heavy menstrual bleeding** | **1.4 (1.0, 1.9)** | **0.04** |
| Extreme level menstrual pain | 1.5 (0.9, 2.5) | 0.17 |
| Socioeconomic quantile (reference poorest) | 1.0 (0.8, 1.2) | 0.75 |
| **Urinary Tract Infection** | | |
| Age≥18y | 1.0 (0.9, 1.4) | 0.93 |
| Married | 1.2 (1.0, 1.5) | 0.06 |
| Maternal education | 1.0 (1.0, 1.01) | 0.34 |
| Less than 30 minutes to collect water at home | 1.0 (0.8, 1.1) | 0.59 |
| **School providing girls with a supply of pads** | **1.4 (1.2, 1.5)** | **<0.005** |
| Frequency of changing pads (reference once a day) | 1.1 (0.9, 1.3) | 0.38 |
| Using other sanitary material other than disposable pads | 0.9 (0.8, 1.1) | 0.40 |
| **Privacy in school toilets** | **0.6 (0.5, 0.7)** | **<0.005** |
| Clean toilets in school | 0.9 (0.9, 1.0) | 0.06 |
| **Water source in school functional 2–4 days per week** | **1.4 (1.3, 1.6)** | **<0.005** |
| Availability of soap to wash hands in toilets | 1.0 (0.8, 1.1) | 0.70 |
| Heavy menstrual bleeding | 1.2 (1.0, 1.4) | 0.08 |
| **Extreme level menstrual pain** | **1.3 (1.2, 1.4)** | **<0.005** |
| Socioeconomic quantile (reference poorest) | 1.0 (1.0, 1.1) | 0.50 |
| **Reproductive Tract Infection** | | |
| Age≥18y | 1.1 (0.9, 1.3) | 0.58 |
| Married | 1.1 (0.7,1.7) | 0.61 |
| **Less than 30 minutes to collect water at home** | **1.2 (1.0, 1.5)** | **0.03** |
| Using other sanitary material other than disposable pads | 1.0 (0.8, 1.1) | 0.23 |
| School providing girls with a supply of pads | 1.0 (0.8, 1.3) | 0.76 |
| Frequency of changing pads (reference once a day) | 0.9 (0.8, 1.0) | 0.23 |
| Privacy in school toilets | 0.8 (0.6, 1.0) | 0.05 |
| Clean toilets in school | 0.9 (0.8, 1.0) | 0.05 |
| Water source in school functional 2–4 days per week | 0.9 (0.8, 1.1) | 0.33 |
| **Availability of soap to wash hands in toilets** | **0.6 (0.5, 0.8)** | **<0.005** |
| **Heavy menstrual bleeding** | **1.4 (1.1, 1.8)** | **0.02** |
| Extreme menstrual pain | 1.2 (1.0, 1.5) | 0.08 |
| Socioeconomic quantile (reference poorest) | 1.0 (0.9, 1.2) | 0.48 |

India [45]. However, when water is available within 1 km, or a 30 min return round trip from the household, water use does not change significantly until water is provided on the plot or very nearby [46]. This could therefore explain why we did not observe any benefit among adolescent girls who took <30 minutes to collect water for the household that was often from a

communal tap in the village. This shows that causality cannot be inferred using this study design and further research is required in rural communities undergoing transition in access to WASH to ascertain the thresholds that have to be met both at home and at school to reduce the risk of RTIs among adolescent girls.

In the context of poverty, the availability of sanitary pads from schools for girls is an important initiative supported by the Ministry of Basic and Secondary Education in The Gambia and NGOs, to improve menstrual hygiene at schools as a means of improving academic performance, improving self-confidence and health as well as reducing drop-out of girls among adolescent girls [3, 47, 48]. The Ministry of Basic and Secondary Education reported preliminary findings from a cross-sectional survey that showed improved school attendance among the girls from 68% pre-supply of sanitary towels to 89.7% post-supply of sanitary towels in the schools that piloted this initiative [48]. However, in our study we found that attending a school that supplied disposable sanitary pads was associated with having UTI symptoms. This could be partly explained by the mismatch between number of disposable pads that were supplied by the schools and the actual numbers that were needed by the girls during their menstrual period. On average, during their last menstrual period each girl reported that they received only one packet with 10 units of disposable pads that would probably have been inadequate particularly for girls who reported heavy menstrual bleeding. Girl with heavy menstrual bleeding presented more RTI symptoms. In our previous study, girls reported that the supply of pads from school was insufficient for the duration of their menstrual periods, and as a result many resorted to changing their pads less frequently, to ensure the supply lasted for longer [3]. Indeed, due to challenges of accessing further supplies after the school supplies had run out, they tended to revert to using reusable absorbents in the form of cloths or towels therefore exposing them to the health risks if optimal hygiene practices are not employed with the reusable material [3]. In addition, the schools were ill-prepared to support the disposal of disposable sanitary pads therefore further limiting the benefits of this initiative.

A previous cross-sectional study in rural Gambia among adult women aged 20-53y found that that bacterial vaginosis was non-significantly more frequent among women who used disposable sanitary pads compared to traditional cloths [49]. In a hospital based survey in India of >500 non-pregnant women aged 18–45 [12], women with *Candida* infection were more likely to use reusable absorbent material (aPRR = 1.54, 95%CI 1.2–2.0) and practice lower frequency of personal washing (aPRR = 1.34, 95%CI 1.07–1.7) [12]. In addition, women with bacterial vaginosis were also more likely to practice personal washing less frequently (aPRR = 1.25, 95%CI 1.0–1.5), change absorbent material outside a toilet facility (aPRR = 1.21, 95%CI 1.0–1.48) whilst a higher frequency of absorbent material changing was protective (aPRR = 0.56, 95%CI 0.4–0.75) [12]. In our study the type of menstrual material used, and other menstrual practices were not associated with symptoms of RTI or UTI. However further work with laboratory diagnosis is required to explore the association between MHM and common RTIs as well as UTIs especially among girls with reported heavy menstrual bleeding in order to support the development of preventative strategies.

Overall the numbers of reported symptoms suggestive of either a UTI (including dysuria, urinary frequency, passing cloudy/blood-stained urine, lower abdominal/vaginal pain) or RTI (including abnormal vaginal discharge, foul/ "fishy" smell from the genital area, burning/itching of genitalia) were high among girls in this population with 75% reporting having at least one UTI symptom in the previous 24 hours and 47% reporting at least one RTI symptom in the previous 2 months. In addition, >50% of girls reported having heavy menstrual bleeding. This is similar to a previous cohort study of >1800 women aged 15-54y in rural Gambia that found that the frequency of reproductive organ morbidity was high, 47% of the women had RTIs (the commonest being bacterial vaginosis that was found in 37%) and 34% had menstrual

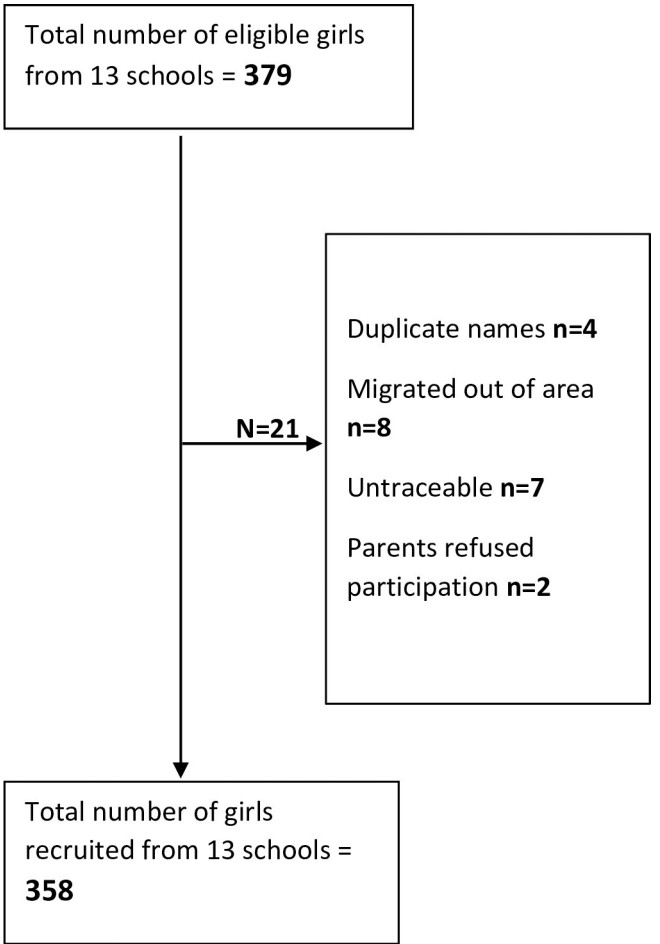

**Fig 1. Flow diagram of the sampling process.**

dysfunction [50]. The commonest reported symptoms were menstrual problems, abnormal vaginal discharge and vaginal itching [50]. In Kenya, a cross-sectional study among adolescents found that 24% had one or more symptoms suggestive of a RTI whilst a third had a confirmed bacterial infection- bacterial vaginosis was the most common [8]. They also found that reported symptoms had a low positive predictive value as most of the girls with confirmed bacterial infections were asymptomatic therefore our study may have underestimated the burden of RTI this population [8]. This has implication for their long-term health and reproductive outcomes. In the cohort study previously mentioned from The Gambia 1 in 10 women were found to suffer with infertility similar to global estimates of secondary infertility, with sub-Saharan Africa having one of the highest rates with previous genital infections being a leading cause [51, 52].

We also found that extreme menstrual pain that was reported by 38% of the girls in our study, was associated with UTI symptoms. Menstrual pain is common among women of reproductive age in The Gambia and is often attributed to female genital mutilation/cutting (FGM/C) (i.e. *partial or total removal of the female external genitalia or other injury to the female genital organs*), which is deeply imbedded in both cultural and religious practices even though it is now prohibited by law [53, 54]. In 2018, over three quarters of adolescent girls aged 15–24 years reported that they had undergone FGM/C, most commonly among the

Mandinka that was the predominant ethnic group in our study [53]. Our findings align with a previous hospital-based cross-sectional study in The Gambia that reported that recurrent symptoms and sign of UTI (14.4% no FGM/C vs 19.6% Type I FGM/C, 44.5% Type II FGM/C, p<0.001) and dysmenorrhea (i.e. painful menstruation) (34.5% vs 58.2% Type I FGM/C, 67.6% Type II FGM/C, p<0.001) were both significantly associated with FGM/C [54]. Although we did not assess the impact of FGM/C on health outcomes in our study, these findings suggest that this may be an important contributor to adolescent adverse health outcomes that are exacerbated by poor MHM practices. This warrants further exploration.

Worryingly, we found that 21% of the girls in our study had symptoms suggestive of depression. Only heavy menstrual bleeding was associated with symptoms suggestive of depression. Although different screening tools have been used for depression in adolescents, the prevalence of symptoms suggestive of depression in our study was similar to that reported among adolescent girls in other sub-Saharan African contexts. A recently published cross-sectional survey in Tanzania of >3000 out of school adolescent girls and young women aged 15-23y using the Patient Health Questionnaire-4 found that 36% of the women had symptoms of depression, and 20% had moderate symptoms [55]. Another cross-sectional study among >500 school going adolescents in rural Uganda aged 14-16y, found that 21% of both the boys and girls had significant depression symptoms on initial screening (using the Children Depression Inventory). In this context being a female (adjusted odds ratio [AOR] 1.50), living in child headed household (AOR 2.20), chronic physical illness (AOR 1.25) and orphan hood (AOR 1.20) were each independently associated with significant depression symptoms [56]. Previous researchers have explained that sex differences are probably due to some combination of age-related changes in biological and social circumstances [57, 58]. A community-based reproductive health survey in rural Gambia among women aged 15–54 years using a modified Edinburgh Depression Scale reported a lower prevalence of depression of 10.3%, but notably only 3% were in the age range for "adolescents" and "young people" [59]. Severe menstrual pain (aOR 3.9) was significantly associated with depression in addition to conditions that would have been more relevant to adult women including divorce/widowhood and infertility [59]. Despite the previously reported stressors associated with menstruation among adolescent girls in this rural Gambian community, including feelings of shame and embarrassment during menstruation and the inadequate access to menstrual absorbents and WASH facilities [3, 48], we did not find an association between specific MHM practices or extreme pain during menstruation and symptoms suggestive of depression. However, we found that girls with heavy menstrual bleeding had more symptoms suggestive of depression, indicating that this subgroup of girls might have extra level of difficulties in managing menstruation than together with other social or personal factors can trigger symptoms suggestive of depression. Further exploration of specific challenges related to menstrual hygiene among girls with heavy menstrual bleeding is required.

## Limitations

This study has a number of limitations. The cross-sectional design of the study mean that our findings may be subject to confounding even after adjusting for covariates in univariable analysis. The health outcomes were primarily based on reported symptoms and a urine dipstick test that were all conducted in the field by well-trained field staff who had close links with the MRC primary health care clinic. However, the girls did not undergo complete clinical assessments with laboratory investigations therefore, we were not able to ascertain clinical diagnoses. Future work in this population should incorporate more comprehensive clinical and microbiological assessments in order to get a better estimate of prevalence of UTI and RTI in this

population, support the development of combined public health and antimicrobial interventions. In addition, we used the BDI II tool to assess for symptoms suggestive of depression among the adolescent but had to drop one question due to the challenges of discussing "interest in sex" with adolescents in this community. Although we opted not to replace or adjust the overall cut-off score for depression as we felt that this was a small change, this may have led to an underestimation of the prevalence of symptoms suggestive of depression among these adolescent girls. In our future work designing adolescent health interventions in this community, we plan to evaluate optimal tools for assessing depression among adolescents that will include adaptations and validation of the modified BDI II.

## Conclusion

Our study found that heavy menstrual bleeding, menstrual pain and having an inappropriate WASH environment are important factors that affect the health of adolescent girls in rural Gambia. Interventions aiming to support schools to design appropriate WASH facilities for appropriate management of menstruation even in the context of limited resources will be a key step in addressing girls´ health. More attention to strategies for pain management and improvement of schools' programmes aiming to distribute sufficient and appropriate menstrual absorbents, particularly among those with heavy menstrual bleeding, will be also required.

## Supporting information

**S1 File.**
(DOCX)

**S1 Dataset.**
(XLSX)

## Acknowledgments

Our sincere gratitude goes out to the West Kiang adolescent girls as well as their parents and teachers for their engagement with this study. We would like to thank the Ministry of Education of the Gambia for their support throughout this study. We would also like to thank Prof Andrew Prentice for inviting us to conduct this research from the MRC The Gambia field station in Keneba. We thank the data management team for their diligence and commitment to ensuring that our data was optimal.

## Author Contributions

**Conceptualization:** Helen M. Nabwera, Vishna Shah, Fatou Sosseh, Wolf-Peter Schmidt, Belen Torondel.

**Data curation:** Vishna Shah, Bakary Sonko, Belen Torondel.

**Formal analysis:** Helen M. Nabwera, Vishna Shah, Belen Torondel.

**Funding acquisition:** Helen M. Nabwera, Vishna Shah, Belen Torondel.

**Investigation:** Belen Torondel.

**Methodology:** Helen M. Nabwera, Vishna Shah, Rowena Neville, Mariama Saidykhan, Fatou Faal, Bakary Sonko, Omar Keita, Wolf-Peter Schmidt, Belen Torondel.

**Project administration:** Vishna Shah, Rowena Neville, Fatou Sosseh, Mariama Saidykhan, Fatou Faal, Bakary Sonko, Omar Keita, Belen Torondel.

**Resources:** Belen Torondel.

**Software:** Belen Torondel.

**Supervision:** Belen Torondel.

**Validation:** Belen Torondel.

**Visualization:** Belen Torondel.

**Writing – original draft:** Helen M. Nabwera, Vishna Shah, Wolf-Peter Schmidt, Belen Torondel.

**Writing – review & editing:** Helen M. Nabwera, Vishna Shah, Rowena Neville, Fatou Sosseh, Mariama Saidykhan, Fatou Faal, Omar Keita, Wolf-Peter Schmidt, Belen Torondel.

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
