## [Decision Letter · Decision Letter 0]

4 Sep 2020

PONE-D-20-15732

Menstrual hygiene management practices and associated health outcomes among school-going adolescents in rural Gambia

PLOS ONE

Dear Dr. Nabwera,

Thank you for submitting your manuscript to PLOS ONE. After careful consideration, we feel that it has merit but does not fully meet PLOS ONE’s publication criteria as it currently stands. Therefore, we invite you to submit a revised version of the manuscript that addresses the points raised during the review process.

First of all, I regret that the reviews have been long. However, I hope and trust that you can use the comments to improve the manuscript and clarify the message.

A marked-up copy of your manuscript that highlights changes made to the original version. You should upload this as a separate file labeled 'Revised Manuscript with Track Changes'.An unmarked version of your revised paper without tracked changes. You should upload this as a separate file labeled 'Manuscript'.

We look forward to receiving your revised manuscript.

Kind regards,

Peter F.W.M. Rosier, M.D. PhD

Academic Editor

PLOS ONE

Journal Requirements:

2. Please include additional information regarding the survey or questionnaire used in the study and ensure that you have provided sufficient details that others could replicate the analyses. For instance, if you developed a questionnaire as part of this study and it is not under a copyright more restrictive than CC-BY, please include a copy, in both the original language and English, as Supporting Information. Moreover, please include more details on how the questionnaire was pre-tested, and whether it was validated.

4. Your ethics statement must appear in the Methods section of your manuscript. If your ethics statement is written in any section besides the Methods, please move it to the Methods section and delete it from any other section. Please also ensure that your ethics statement is included in your manuscript, as the ethics section of your online submission will not be published alongside your manuscript.

Reviewers' comments:

Reviewer's Responses to Questions

**Comments to the Author**

1. Is the manuscript technically sound, and do the data support the conclusions?

Reviewer #1: Partly

Reviewer #2: Yes

2. Has the statistical analysis been performed appropriately and rigorously? 

Reviewer #1: Yes

Reviewer #2: Yes

3. Have the authors made all data underlying the findings in their manuscript fully available?

Reviewer #1: Yes

Reviewer #2: Yes

4. Is the manuscript presented in an intelligible fashion and written in standard English?

Reviewer #1: Yes

Reviewer #2: Yes

5. Review Comments to the Author

Reviewer #1: Overall a good topic to address, and needed research.

However conceptually it is hard to follow as there are too many outcomes and no focused message as a result. The results are also simply described and plausibility of associations are not explained. I believe a major revision is needed.

I recommend to refine the study by focusing on the messages in the concluding paragraph and then presenting data that supports (or not) those objectives and explaining them more in depth. Much of the data presented could be removed in order to have a more coherent message.

Technically, it is unclear if the study instruments for data collection were validated. If so, please describe or reference.

Model selection process is not explained, nor are the co-variables.

Please report the rate of missing data in the final model.

Reviewer #2: Questions for the authors from the reviewer

1. Aims and objectives of the study were not stated clearly in the manuscript?

2. Using an estimated rate of prevalence of urogenital infection of 27%, use of reusable cloth of 40%, prevalence of drying inside the house of 21.6% and prevalence of school absenteeism of 23%. Based on these estimates and using precision/absolute error of 5% and a type 1 error of 5%, a sample size of 368 was calculated. – Which one of the estimate was used in calculating the sample size? Explain the calculation in the methodology?

3. Girls who were pregnant were excluded – How was it ensured?

4. Standard pretested questionnaire to collect sociodemographic data and quantify knowledge and practices of menstruation. – Is the MHM questionnaire used in this study, the standardized and pre validated questionnaire? Or whether the Questionnaire was pretested and validated by the research team of this paper?

5. In addition, a modified Beck Depression Inventory II (BDI II) was used to screen for symptoms of depression among the adolescents. BDI II has 21 items with a score range of 0-3 and is a validated screening tool for depression in both adults and adolescents worldwide [27, 28]. A score of > 16 is suggestive of clinical depression [29]. We dropped one item that explore “interest in sex” from our study as this was not culturally appropriate for this setting. Our modified version therefore had 20 items. – What was the validation process adopted after deleting one question from a validated inventory scale? How was the scoring done for depression after deleting one question?

6. Mid-stream urine samples were collected for urine dipstick analysis using the Combur-test strips [30] that was performed at the schools. What’s purpose of Combur test strip? – Explain the standard guidelines, protocols and procedures followed for sample collection and the performance of the test in the methodology?

7. A pre-determined tool was used to assess the WASH hardware available; number and type of toilets, handwashing facilities and disposal facilities. As well as access the quality of these facilities; functionality, gender specificity, privacy, cleanliness, and availability of water and soap. Did the research team informed to the school authorities about these inspection / supervising activities on the facilities beforehand? If yes – How early it was informed to them?

6. PLOS authors have the option to publish the peer review history of their article (what does this mean?). If published, this will include your full peer review and any attached files.

Reviewer #1: No

Reviewer #2: **Yes: **Dr. Balaji Arumugam

---

## [Author Response · Author response to Decision Letter 0]

15 Oct 2020

15th October 2020

Peter F.W.M. Rosier, M.D. PhD

Academic Editor

PLOS ONE

Dear Dr Rosier,

RE: Manuscript reference number: PONE-D-20-15732

Title: “Menstrual hygiene management practices and associated health outcomes among school-going adolescents in rural Gambia."

Thank you for your e-mail from the 4th September 2020, requesting us to revise our manuscript to incorporate the editorial and reviewers’ comments. Please find below a point-by-point response to all yours and your reviewers’ comments. We have indicated where changes have been made in the text, using page and line numbers of the “tracked changes” version. Also find uploaded the revised manuscript and supplementary material. We have also uploaded a clean copy. I trust that we have addressed all your comments adequately and that you would kindly consider publishing our manuscript in your journal.

Editorial comments: 

There are no ethical or legal restrictions on sharing a de-identified data set. Data access requests can be sent to Dr Belen Torondel Belen.Torondel@lshtm.ac.uk

Data will be available as Supporting Information file. 

 Thank you.

4. Your ethics statement must appear in the Methods section of your manuscript. If your ethics statement is written in any section besides the Methods, please move it to the Methods section and delete it from any other section. Please also ensure that your ethics statement is included in your manuscript, as the ethics section of your online submission will not be published alongside your manuscript.

Authors: The ethics statement information is already in the Method section. 

Reviewers’ comments:

Reviewer #1: Overall a good topic to address and needed research.

1. However conceptually it is hard to follow as there are too many outcomes and no focused message as a result. The results are also simply described and plausibility of associations are not explained. I believe a major revision is needed.

I recommend to refine the study by focusing on the messages in the concluding paragraph and then presenting data that supports (or not) those objectives and explaining them more in depth. Much of the data presented could be removed in order to have a more coherent message.

Authors: We understand the concerns of the reviewer, however we believe it is important to present all the outcome data available in this study, as there is a scarcity of information about this topic in Gambia. Our study was exploratory, seeking to describe adverse health outcomes associated with poor menstrual hygiene management among adolescent girls in a rural context of The Gambia. We therefore focussed on 3 main conditions that the literature suggested were associated with poor menstrual hygiene management in other low-and middle-income countries i.e. urinary tract infections (using common symptoms and urine dipstick); reproductive tract infections (common symptoms) and depression (modified Beck depression inventory). We appreciate your comments, and they have been very useful to help us revise the manuscript and clarify which outcomes we are studying along the different manuscript sections (Abstract, objectives, methods). We have also rewritten some paragraphs of the results and discussion section and provided more clear explanation of plausibility of associations of the different risk factors identified and the different outcomes. Hope the manuscript is clearer now. 

2. Technically, it is unclear if the study instruments for data collection were validated. If so, please describe or reference.

Authors: Thanks for your comment, and to notice that we did not include information about how we validated our data collection tools. We have added a small sub-section explaining how we tested and validated our study instruments (Page 7, lines 7-25; Page 8, lines 1-5).

3. Model selection process is not explained, nor are the co-variables.

Authors: In page 8, line 7-13 we have described the model selection process in more detail and provided more information on where the details of the co-variables for each of the 3 multivariable models can be found. There was one model for each of the three health outcomes ((i.e. depression, UTI, RTI). For each model explanatory variables that were statistically significant in the univariable analysis as well as those that had biologically plausible associations with each of the three health outcomes were included. 

4. Please report the rate of missing data in the final model.

Authors: Rates of missing data have been included in the footnote of Tables 1 (Sociodemographic) & Table 3 (Menstrual Hygiene Management). We have also included the details of the missing data in each of the models. (Page 9, line 22-24) 

Reviewer #2: Questions for the authors from the reviewer

1. Aims and objectives of the study were not stated clearly in the manuscript?

Authors: Thanks for your comment, we have added a clearer aim and objective of the study at the end of the introduction section, hope now the objectives are clear. (Page 4, lines 1-6)

2. Using an estimated rate of prevalence of urogenital infection of 27%, use of reusable cloth of 40%, prevalence of drying inside the house of 21.6% and prevalence of school absenteeism of 23%. Based on these estimates and using precision/absolute error of 5% and a type 1 error of 5%, a sample size of 368 was calculated. – Which one of the estimates was used in calculating the sample size? Explain the calculation in the methodology?

Authors: Thanks for your comment, we forgot to include that we used 40% prevalence estimated as this resulted in the larger sample size. We have included clarification in the sample size section. 

3. Girls who were pregnant were excluded – How was it ensured?

Authors: Thank you for your comment. During participant recruitment the field assistants asked the girls if they were pregnant. If they said “yes” then they were excluded from the study. We have added these details to the Methods section. (Page 5, lines 19-20)

4. Standard pretested questionnaire to collect sociodemographic data and quantify knowledge and practices of menstruation. – Is the MHM questionnaire used in this study, the standardized and pre validated questionnaire? Or whether the Questionnaire was pretested and validated by the research team of this paper?

Authors: Thank you for your comment. All the tools that we used, were validated previously in other studies, however we adapted them to our local context (when needed) and we also piloted them before collecting the final study data. We have added more information in the Methods section, explaining how we develop the tools, how we tested and how we collected the study data. (Page 7, lines 7-25; Page 8, lines 1-5)

5. In addition, a modified Beck Depression Inventory II (BDI II) was used to screen for symptoms of depression among the adolescents. BDI II has 21 items with a score range of 0-3 and is a validated screening tool for depression in both adults and adolescents worldwide [27, 28]. A score of > 16 is suggestive of clinical depression [29]. We dropped one item that explore “interest in sex” from our study as this was not culturally appropriate for this setting. Our modified version therefore had 20 items. – What was the validation process adopted after deleting one question from a validated inventory scale? How was the scoring done for depression after deleting one question?

Authors: Thank you. We have added more information in the Method section about the process of adapting the Beck Depression Inventory II (BDI II) to this rural Gambian context. Discussing “interest in sex” among adolescents in this community was a taboo in this community but as this tool has been validated for use in sub-Saharan Africa, we were keen to use it in this exploratory study. (Page 7, lines 11-25) We viewed dropping this one question as a small change that did not require validation, but we appreciate that this approach may limit the generalisability of our findings. We have discussed this under the limitations. (Page 21, lines 9-16) 

6. Mid-stream urine samples were collected for urine dipstick analysis using the Combur-test strips [30] that was performed at the schools. What’s purpose of Combur test strip? – Explain the standard guidelines, protocols and procedures followed for sample collection and the performance of the test in the methodology?

Authors: Thank you. The Combur test strips were used to test the urine for any evidence of a urinary tract infection. We have added the requested information in the methods section and as supplementary material. (Page 6, lines 10-22)

7. A pre-determined tool was used to assess the WASH hardware available; number and type of toilets, handwashing facilities and disposal facilities. As well as access the quality of these facilities; functionality, gender specificity, privacy, cleanliness, and availability of water and soap. Did the research team informed to the school authorities about these inspection / supervising activities on the facilities beforehand? If yes – How early it was informed to them?

Authors: Thank you for your comment, one unannounced visit to conduct spot check in each school was conducted. Permissions to conduct unannounced visits were obtained from each school at the beginning of the study. We have added this information in the method section. (Page 6, lines 25-26; Page 7, lines 1-4) 

Kind regards, 

Belen Torondel 

Department of Disease Control, Faculty of Infectious Diseases

London School of Hygiene and Tropical Medicine

Keppel St, London WC1E7HT, UK

+44 (0) 20 7636 2934

---

## [Decision Letter · Decision Letter 1]

26 Jan 2021

PONE-D-20-15732R1

Menstrual hygiene management practices and associated health outcomes among school-going adolescents in rural Gambia

PLOS ONE

Dear Dr. Nabwera,

Thank you for submitting your manuscript to PLOS ONE. After careful consideration, we feel that it has merit but does not fully meet PLOS ONE’s publication criteria as it currently stands. Therefore, we invite you to submit a revised version of the manuscript that addresses the points raised during the review process.

ACADEMIC EDITOR:

I only have one small request before I start advising publication: Although the BDI is well validated, also for screening, it cannot be used to diagnose depression as a clinical psychiatric diagnosis, especially the socio-economic hopelessness, which of course is inextricably linked with emotional well-being, will probably also or partly lead to a high score. I agree that you mention that a proportion of respondents has a score that matches depression. But I ask that you consider modifying the phrases where you state that these individuals 'have depression'.

We look forward to receiving your revised manuscript.

Kind regards,

Peter F.W.M. Rosier, M.D. PhD

Academic Editor

PLOS ONE

Reviewers' comments:

Reviewer's Responses to Questions

**Comments to the Author**

1. If the authors have adequately addressed your comments raised in a previous round of review and you feel that this manuscript is now acceptable for publication, you may indicate that here to bypass the “Comments to the Author” section, enter your conflict of interest statement in the “Confidential to Editor” section, and submit your "Accept" recommendation.

Reviewer #2: All comments have been addressed

2. Is the manuscript technically sound, and do the data support the conclusions?

Reviewer #2: Yes

3. Has the statistical analysis been performed appropriately and rigorously? 

Reviewer #2: Yes

4. Have the authors made all data underlying the findings in their manuscript fully available?

Reviewer #2: Yes

5. Is the manuscript presented in an intelligible fashion and written in standard English?

Reviewer #2: Yes

6. Review Comments to the Author

Reviewer #2: Its a good public health important study, kindly address the study population regarding health education on menstrual hygiene practices utilizing the peer sessions among them which I had practiced in our field practice area in schools which gave god long term positive outcomes.

7. PLOS authors have the option to publish the peer review history of their article (what does this mean?). If published, this will include your full peer review and any attached files.

Reviewer #2: **Yes: **Dr. Balaji Arumugam

---

## [Author Response · Author response to Decision Letter 1]

6 Feb 2021

Thank you for this comment. We have revised the phrases referring to "depression" in the manuscript.

---

## [Editor Report · Decision Letter 2]

10 Feb 2021

Menstrual hygiene management practices and associated health outcomes among school-going adolescents in rural Gambia

PONE-D-20-15732R2

Dear Dr. Nabwera,

We’re pleased to inform you that your manuscript has been judged scientifically suitable for publication and will be formally accepted for publication once it meets all outstanding technical requirements.

Kind regards,

Peter F.W.M. Rosier, M.D. PhD

Academic Editor

PLOS ONE

Additional Editor Comments (optional):

none
---

## [Editor Report · Acceptance letter]

15 Feb 2021

PONE-D-20-15732R2 

Menstrual hygiene management practices and associated health outcomes among school-going adolescents in rural Gambia 

Dear Dr. Nabwera:

I'm pleased to inform you that your manuscript has been deemed suitable for publication in PLOS ONE. Congratulations! Your manuscript is now with our production department. 

Kind regards, 

on behalf of

Dr. Peter F.W.M. Rosier 

Academic Editor

PLOS ONE